# Analysis of lithium trade patterns and influencing factors in the regions along the "Belt and Road"

**Ziyun Ruan, Peng Du***, Yanan Jiao

School of Geography, Liaoning Normal University, Dalian, China

* dpdl1984@163.com

## Abstract

Lithium has broad applications in several emerging industries and fields, including high energy batteries, energy storage, aerospace, and controlled nuclear reactions. Currently, the discrepancy between the supply and demand for lithium resources increases, and its distribution is uneven. Within the framework of the "Belt and Road" Initiative, the lithium trade pattern evolves constantly. However, the trade pattern of lithium in the nations along the "Belt and Road" is likely to face substantial repercussions in modern world of unilateral protectionism and geopolitical conflicts. Taking the social network analysis approach as a tool, this study first examines the characteristics of the lithium trade network structure as it has evolved over the years in the Belt and Road countries, from 2000 to 2022. Additionally, this study uses the quadratic assignment problem approach to analyze the factors influencing the evolution of the lithium trade network. The study shows that: (1) The spatial patterns of import and export trade network of lithium in countries along the route has a certain path dependence. And the market is mainly concentrated in East Asia, Central and Eastern Europe, South America and Southeast Asia. (2) The network density of the countries along the route has increased year after year, but it remains low. And the fluctuation of the network's reciprocity has increased, with a huge magnitude of variation. The number of core countries in the network has decreased over time, but the core-periphery structure has stayed largely steady. China, Chile, and South Korea are the network's main node countries. (3) Regarding the influencing factors, the differences in economic and technological development between these countries have a beneficial impact on the formation of lithium trade; whether or not regional trade agreements have been signed, the differences about average tax rates for mineral products, bordering countries, and similar languages and cultures are all conducive to the establishment of close trade links. The contribution of this essay is of paramount importance for understanding different countries' role along the Belt and Road in the lithium trade network pattern, and promoting regional trade cooperation.

## 1 Introduction

In 2013, Chinese President Xi Jinping proposed the Belt and Road Initiative (BRI); and in 2023, the third Belt and Road Summit on International Cooperation has been held in Beijing.

**Data Availability Statement:** All relevant data are within the manuscript and its Supporting information files.

**Funding:** The authors are very grateful for the help of Assistant Professor Peng Du and the support of the National Natural Science Foundation of China, grant number 41701123. The funder role including Conceptualization and writing review & editing.

**Competing interests:** The authors have declared that no competing interests exist.

The Belt and Road Cooperation Initiative, a model of South-South cooperation, has produced impressive results in the last ten years in the fields of energy, commerce and economy, building infrastructure, and green development. It has also developed into a significant global public utility and useful platform that has aided in global development and has kept moving in the direction of sustainability, high standards, and improving people's quality of life [1]. In light of the intricate and significant changes occurring in the world today, including the escalation of protectionist inclinations and the persistent rise in uncertainty regarding economic globalization, China has offered a Chinese solution to the global governance system by fostering greater collaboration among the vast majority of developing nations, advancing the economic growth and infrastructure building of the nations along the Belt and Road, and fostering friendship and prosperity. Considering the intricate and significant changes taking place in the contemporary world, the BRI is a major initiative for China to integrate into the Eurasian continent in the new era, and the pattern of energy power shift that characterizes the Belt and Road region is destined to raise major issues [2]. The unstable pattern of international trade and the uneven distribution of key mineral resources resulted in a complex trading dynamic between the nations involved in the Belt and Road, which is not amenable to easy definition and description, forming a complex network structure [3]. In addition, the trading network has a "hybrid" structural framework that exhibits a link design consisting of both hub-and-spoke and fully connected configurations, resulting in high levels of vulnerability and robustness [4].

Regarding social network research, Freeman [5] systematically described the history of the development of social network analysis methods, and argued that many of the concepts in social network theory originated in graph theory, where a set of points consisting of many points and their corresponding lines or edges connecting the points to each other can be depicted on a blank piece of graph paper. Subsequently, Newman [6] found that networks are usually not random and that they have very distinct statistical features, some of which, such as high clustering coefficients and highly skewed degree distributions, are common in networks of all types. Then the world trade network occurred, as discovered by Serrano [7], is a complex network that differs greatly from a traditional random network because of the vertices' degree-correlation features, small-world traits, and prevalence of scale-free distributions. Garlaschelli [8] first defined the world trade network as a network defined by trade relations among countries in the world. Fagiolo [9] used a weighted network approach to study the empirical properties of the network of trade relations among countries and its evolution over time, and found that most countries have weak trade links, but some countries have strong links. Alexandre [10] used a link prediction algorithm based on a current snapshot of the network aimed at predicting its future evolution, and applied it to the international trade data among countries to obtain better results. Ashadun Nobi et al. constructed a minimum spanning tree in world trade based on cross-correlations between time series of different commodities and found that China's influence on the trade network began to emerge after 2004 [11]. In recent years, the social network analysis method is widely used in international trade and other fields, and the material, information and spatial flows of international trade are one of the main research contents of world economic geography. Scholars at home and abroad apply this method in the study of trade networks in different fields and commodities, such as Guo Weidong [12] studied the evolution of the spatio-temporal pattern and driving mechanism of global military science and technology trade in 3 aspects, namely, product structure, network pattern and influence mechanism. Qi Wei et al. [13] selected the 2017–2020 global new energy automobile trade data as a sample to study the dynamic evolution characteristics of the trade network. Xiaoliang Jia et al. [14] proposed a novel wavelet-based complex network method, aim to study the evolution characteristics of the integration and diversification for the world's crude oil market, from the structural perspective of global oil prices' interdependence. He Shengbing

et al. [15] used the 54 environmental products' trade data of the "Belt and Road" countries from 2000 to 2020 to constructed an environmental product trade network, and used the time-indexed random graph model to explore the structural characteristics of the regional environmental product trade network and its evolution mechanism. In addition, some scholars have also utilized social network theory to construct the trade network of nickel ore [16], chromium ore [17] and licorice extract [18] to analyze their trade structure and trade competitiveness.

Regarding the research on lithium, lithium is the lightest metal in nature, with excellent performance, characterized by strong ability to gain and lose electrons and high specific capacity, which is widely used in a number of new technology industries and fields, such as high-energy batteries, energy storage, aerospace, controlled nuclear reactions, and so on [19, 20]. Currently, the status of lithium in the international arena is rising, and major economies have increased the importance of mineral resources such as lithium. In 2009, the Japanese government first released the Rare Metals Safeguard Strategy [21]; in 2016, China released the National Mineral Resources Plan (2016–2020), which lists 24 minerals (groups) such as oil, natural gas, and other minerals as strategic minerals [22]; in 2022, the U.S. Geological Survey published a new list of 50 key minerals, and the Australian government released the 2022 Key Minerals Strategy, with lithium included in the above strategic list [23, 24]. However, lithium has a certain economic vulnerability, and its industrial chain can be mainly divided into four levels, upstream products (lithium raw ore), basic lithium products, deep-processed lithium products and downstream products [25]. In recent years, the social network analysis method has also been introduced in lithium resources and products trade research. Chen Guang [26] analyzed the characteristics of global lithium carbonate and lithium hydroxide trade communities' evolution. Zhu Lili et al. [27, 28] analyzed the control power of China in the international trade network of lithium hydroxide and lithium carbonate. Wu Qiaosheng et al. [29] established a global dynamic trade network for lithium products, analyzing the degree of trade diversity and trade influence of China in the global trade network through weighted degree and other indicators. However, most of the above studies focus on the community division of lithium trade or the controlling role of a certain country in the trade network from the global or China's perspective, and less on the lithium trade pattern evolution and its influencing factors from the Belt and Road perspective, which makes it difficult for the countries along the route to study their relative position and influence in the lithium trade network.

In conclusion, there are currently more abundant international studies related to lithium, and there are also many studies on trade networks. However, some studies have utilized the intensity of access to identify important nodes, but have not taken into account the importance of the nodes associated with them in the trade network. In view of this, this study not only portrays and analyzes the long time series dynamic network evolution of lithium in the countries along the Belt and Road and its influencing factors, but also identifies the key nodes in the trade network of the Belt and Road by using the PageRank (PR) algorithm. The PR algorithm is used to identify key nodes in the "Belt and Road" trade network, and the PR algorithm can rank the search results to prioritize the more important and core web pages or nodes [30]. Unlike the existing literature that uses access degree to identify important nodes in the network, this study attempts to combine the centrality, core-edge structure, and more advanced PR algorithms to explore the evolution of the spatio-temporal pattern of lithium trade in the perspective of the Belt and Road, to analyze the key nodes in the network, and to identify the key nodes in the "Belt and Road" trade network among the countries along the routes. The role and status of the countries along the route in the "Belt and Road" lithium trade network, in order to provide a scientific basis for the selection of trade partners of the countries along the route. It also analyzes the influencing factors of the trade network evolution from five

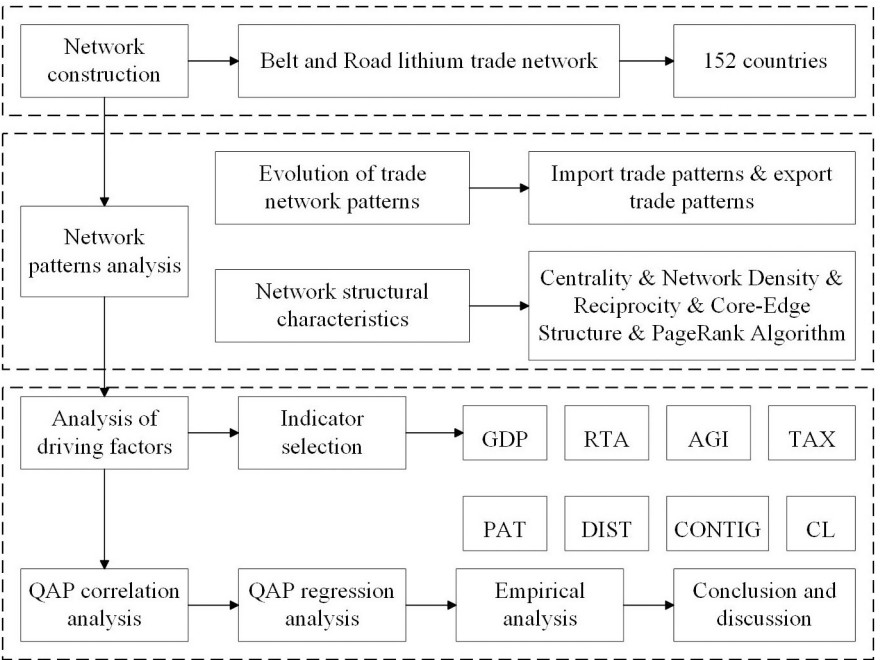

**Fig 1. Flowchart of the main content.**

perspectives: economy, system, science and technology, geography and culture, with a view to optimizing the energy trade strategies and policies under the framework of "Belt and Road", and guaranteeing the high-quality development of China's new energy and new technology industries. The main contents are shown in Fig 1.

## 2 Study area, data sources and research methodology

### 2.1 Study area

The Belt and Road (B&R) is the acronym of the "Silk Road Economic Belt" and the"21st Century Maritime Silk Road". Since its launch, 152 countries or regions have joined and signed cooperation agreements, indicating a continuous and dynamic development process. For the continuity and comparability of academic research, some contemporary scholars take the traditional 65 nations within the B&R as the object of research [31]. However, given the rampant prevalence of unilateral protectionism is rampant in today's world, and the subsequent frustration of the multilateral trading system, the B&R serves as a significant initiative towards creating a community of human destiny. It is an inclusive and open global network for regional economic collaboration [32], which provides a novel boost towards regional economic growth and achieving mutual prosperity. This research investigates the 152 countries encompassed within the B&R, including China.

This paper's time span of 2000 to 2022 was chosen for its ability to highlight the long time series' evolutionary features. The year 2000 is the starting year of this study; in 2003, the world economy became extremely fragile and unstable due to the SARS epidemic and currency wars; in 2008, it witnessed the commencement of the world financial crisis, and then the global economic development entered a downturn; in 2013, China put forward the BRI, which profoundly altered the patterns of development for trade and economics worldwide as well as the

worldwide allocation of labor [33]; in 2019, the outbreak of the COVID-19 epidemic, a public health event, had tremendous impacts and challenges for countries around the world; in 2022, the Russia-Ukraine conflict had an unprecedented geopolitical impact on the B&R. In light of this, this study selects six representative years, specifically 2000, 2003, 2008, 2013, 2019 and 2022, as the cross-section sample. This decision aims to showcase the evolution of basic lithium product trade development in the B&R and the significant turning points.

## 2.2 Data sources and processing

This essay utilizes import and export data from the UN Comtrade Database (https://comtradeplus.un.org/) for lithium, such as lithium hydroxide and lithium oxides, lithium carbonate and lithium perfluorooctyl sulfonate, from nations and areas across the B&R between 2000 and 2022. Considering that the trade volume of Lithium perfluorooctyl sulfonate is too negligible, it has been excluded, and the total number of data points obtained is 11,377. After verifying the customs HS code, the definition of specific lithium and its corresponding customs code are shown in Table 1.

## 2.3 Research methodology

The social network is essentially composed of nodes and edges, and it represents a collection of social players and their connections. This study primarily selects the data of import countries, export countries and trade value, and takes the lithium trade nations throughout the B&R as nodes, as well as the global flows of trade within the route as edges, respectively, to construct the lithium trade network of oriented weighted and unoriented unweighted from 2000 to 2022, which can be summarized as follows in the abstract:

$$G = (V, E, W_i, W_j) \tag{1}$$

Where: V is the collection of all trade network nodes (countries), E is the collection of all trade network edges (links between trading countries), $W_i$ means the trade volume weight matrix in the trade network, and $W_j$ represents the unweighted 0–1 matrix in the trade network.

In this essay, network density, core-edge structure, and block model in social network analysis are used to present the overall characteristics of the Belt and Road lithium trade network, and the node strength is measured by centrality and PR algorithm to portray the characteristics of different nodes in the trade network, and the meanings of each measure and the calculation formulas are shown below.

**2.3.1 Network density and reciprocity.** Network density reflects the extent of connectivity between vertices in a network, and trade network density is an indicator of the closeness of trade linkages between nations. The higher the trade network density, the stronger the connectivity between nations that participate in the network. Conversely, the fewer commercial connections there are among nations. Reciprocity is defined as the fraction of relationships in the structure that allow for two-way trade to the total number of relationships established in the

**Table 1. Classification and definition of customs codes for lithium.**

| Customs codes | Product definitions |
|---|---|
| 282520 | Lithium hydroxide and lithium oxides |
| 283691 | Lithium carbonate |
| 290433 | Lithium perfluorooctyl sulfonate |

Note: Data from China Customs.

network, which measures the degree of connectedness between the two nodes in the network, can be used to determine how stable the two-way commerce in lithium has become. The more reciprocity exists, the more stable the trade network structure of basic lithium products becomes. Network density can be calculated using the following formula:

$$D = \frac{2M}{N(N-1)} \tag{2}$$

Where: M is the number of edges, and N is the number of nodes D is the network's density. The closer D is to 1, the higher the network density and the closer the trade relationship between nations; the closer D is to 0, the sparser the network and the looser the structure of the trade network between nations.

**2.3.2 Core periphery structure.** The trade network's nodes that are in the edge position and those that are in the core position can be identified by core periphery structure research [34]. The UCINET software calculates the core degree for each nation that takes part in the B&R trade network for lithium. The core degree is subsequently utilized as a criterion to identify the core and peripheral countries, which are visualized by the Gephi software.

**2.3.3 Block model.** The block model studies the distribution of network segments and the attribute characteristics of each segment. Referring to the research method of Zhong Zhaohui [35] and Qi Wei [13], the n countries in the network are divided into 4 section (Table 2), and the number of countries within each section is $n_k$.

Where, $(n_k - 1)/(n - 1)$ for the segment k internal trade volume accounted for segment k in the trade network in the total trade volume proportion of the expected value; $I_k/I_{k,t}$ for the plate k internal trade volume accounted for plate k in the network of the total trade volume proportion of the plate k, the plate within the proportion of trade volume higher than the expected value, then the country and the other countries within the plate trade links closer; $I_{ke}/I_{ki}$ for the plate k to the other plate exports to the imports ratio, the ratio is higher than 1, then the countries within the plate to the is the ratio of exports to imports of plate k to other plates, a ratio higher than 1 indicates that the country in the plate has strong exports to other plates; $I_k/I_{k,t} \geq (n_k - 1)/(n - 1)$ and $I_{ke}/I_{ki} \geq 1$ is a two-way trade plate, where the country in the plate has strong trade links with both inside and outside the plate; and is similar to a hole in the structure of a social network, being an isolated marginal plate where the strength of both internal and external trade links is relatively small.

**2.3.4 Centrality.** *(1) Degree centrality.* It is a measure of the size of a node's degree. It can intuitively reflect the size of the node's likelihood of having direct contact with its neighbors in the system. The likelihood of establishing direct communication with other nodes increases with the node's degree centrality rating. The formula is:

$$DC_i = \frac{K_i}{n-1} \tag{3}$$

where: $K_i$ indicates the total quantity of edges that are currently connected to node i, and n-1 denotes the entire quantity of edges where node i is connected to all other nodes.

**Table 2. Block classification method of block model.**

| Proportion of intra-sectoral trade relations | Ratio of trade volume of exports and imports of the sector to other sectors | |
|---|---|---|
| | $I_{k,e}/I_{k,i} \geq 1$ | $I_{k,e}/I_{k,i} < 1$ |
| $I_k/I_{k,t} \geq (n_k - 1)/(n - 1)$ | Two-way trade sector | Import trade sector |
| $I_k/I_{k,t} < (n_k - 1)/(n - 1)$ | Export trade sector | Isolated marginal sector |

*(2) Betweenness centrality.* Measured by counting how many times a specific node appears on the shortest path connecting all nodes, which reflects the node's role as a bridge and transit in the system. The larger the value of intermediary centrality of a node, the greater the connectivity and access of the node. The calculation shown is:

$$BC_k = \frac{2}{n^2 - 3n = 2} \sum_{i=1, j \neq k}^{n} \sum_{j > i, j \neq k}^{n} \frac{S_{ij}^k}{S_{ij}}$$  (4)

Where $S_{ij}$ denotes the total number of shortest distances, $d_{ij}$ from node i to node j, and $S_{ij}^k$ is the number of shortest paths among these $S_{ij}$ shortest paths that go through node k.

*(3) Closeness centrality.* Measured by the size that constitutes shortest distance summation from a given node to all nodes, reflecting the relative accessibility size of the node in the network. Ceteris paribus, nodes with greater closeness centrality are more vital and influential in the network. The equation is:

$$CC_i = \left[ \frac{1}{n-1} \sum_{j=1, j \neq 1}^{1} d_{ij} \right]^{-1}$$  (5)

Where: n remains the overall number of nodes and $d_{ij}$ is the length of shortest path connecting node i to node j.

**2.3.5 PageRank algorithm.** The application of a Markov chain model of random walk on a directed graph is the central concept of the PageRank algorithm. This model explains how a random walker moves randomly along the edges of the graph, visiting from one node to another. Subject to certain conditions, this random wandering process eventually converges to a smooth distribution [34]. In this smooth distribution, the probability of each node being visited is its PageRank value, which can be considered as a measure of the importance about the node. The equation for this distribution is as follows:

$$PR(a)_{i+1} = \sum_{i=0}^{n} \frac{PR(Ti)_i}{L(Ti)}$$  (6)

Where: i stands for the present instant or iteration number; PR(a) denotes the PR value of the current node a; PR(Ti) indicates the PR value of each other node (able to point to a); L(Ti) denotes the count of outgoing chains of each other node (able to point to a).

**2.3.6 Quadratic assignment procedure method.** The QAP method, or Quadratic Assignment Procedure, captures the correlation and regression between relational data. It is based on the substitution of matrix data and gives the correlation coefficients between two matrices by comparing the grid values corresponding to each square matrix, while the coefficients are subjected to nonparametric tests [36]. QAP regression, on the other hand, examines the regression relationship between multiple factors or matrices and a matrix, and evaluates the significance of the coefficient of determination $R^2$, which is a good method that can rule out spurious structural relationships. When performing QAP multivariate regression, all matrices must be N x N square matrices. This approach reduces the problem of a multivariate partial regression coefficient to a simple regression where the bias of the regression analysis does not increase with autocorrelation. In this bivariate form, QAP can assess whether the correlation is significantly different from zero and whether the corresponding multiple regression is significantly different from zero [37].

## 3 Analysis of the basic lithium products' trade network pattern in countries along the Belt and Road

### 3.1 Analysis of overall import trade pattern

Figs 2 and 3 show the trade network patterns of import and export of lithium in nations across the B&R in 2000, 2003, 2008, 2013, 2019 and 2022, respectively. The connecting

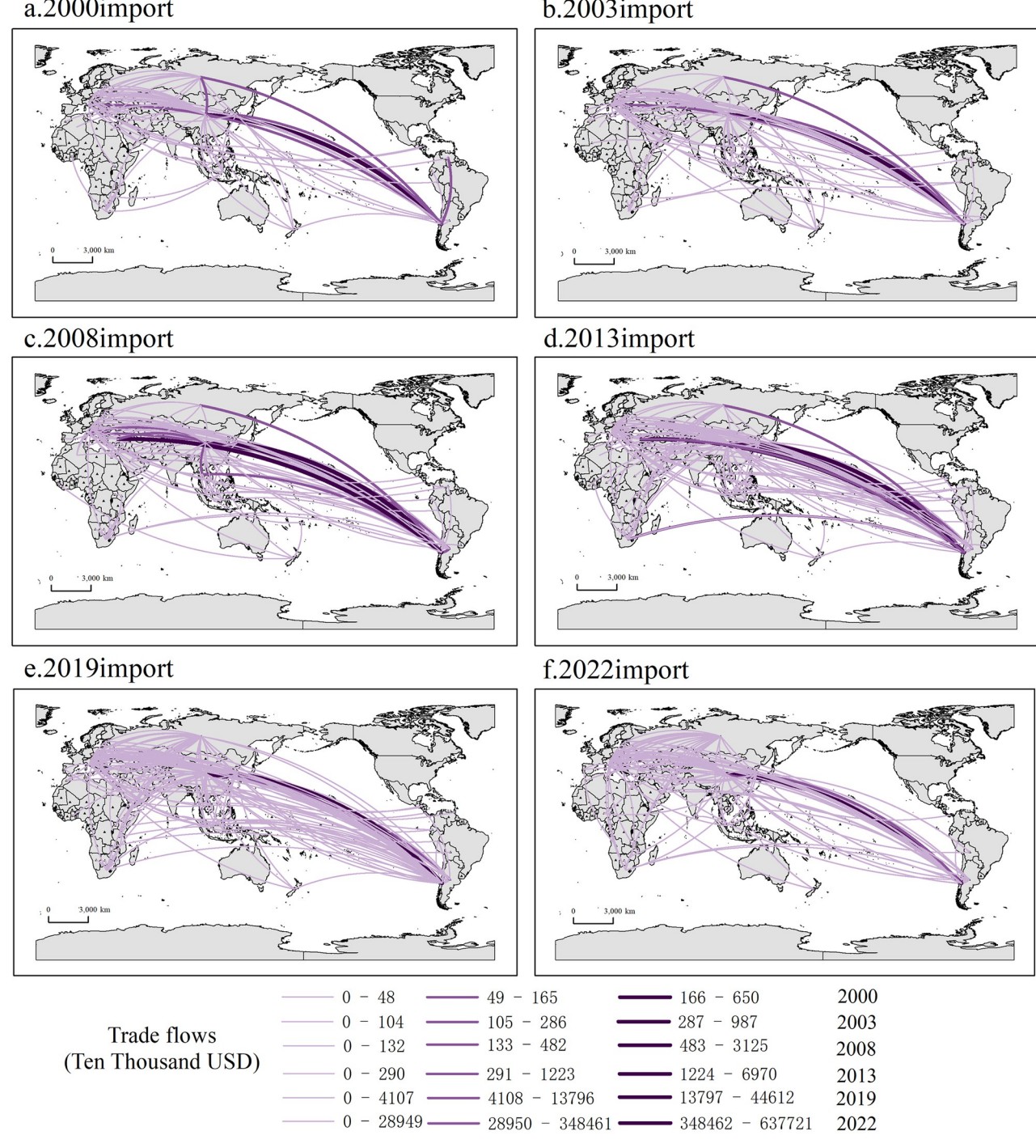

**Fig 2. Import pattern of lithium of countries along the "Belt and Road".**

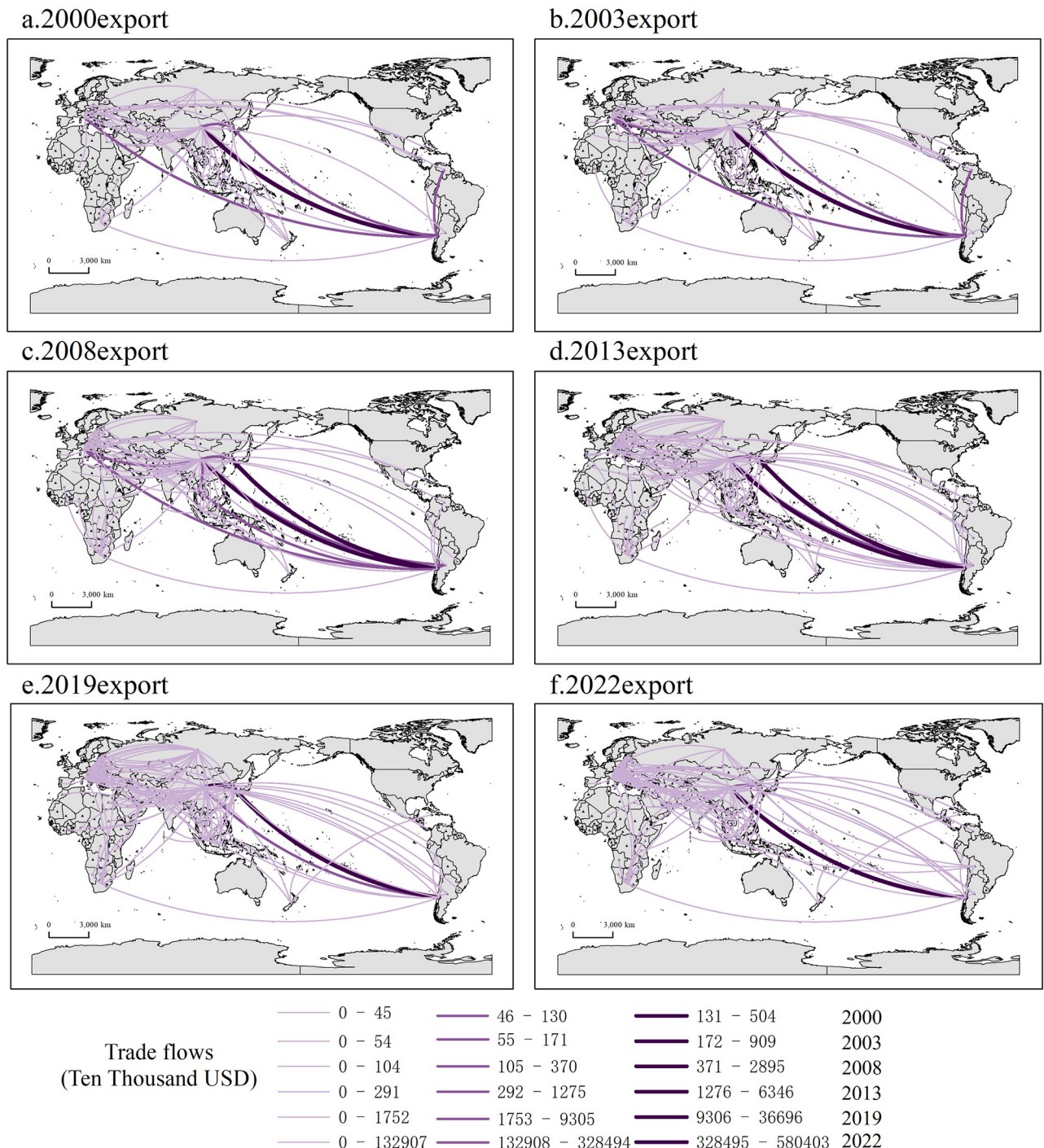

**Fig 3. Lithium's export pattern of countries along the B&I.** Note: Figs 2 and 4 were produced based on the standard map GS (2016) 1667 on the website of the Standard Map Service of the Ministry of Natural Resources of China, with no modifications to the boundaries of the base map.

lines depict the lithium trade relations between two countries in a particular year. The more connecting lines present, the greater the number of trade exchanges between nations. And the darker the color of the linking lines, the higher the trade volume between countries.

From the import perspective, the import pattern of lithium in the countries along the route is highly concentrated, with an obvious pathway dependence. The countries along the trail's import markets during the period 2000–2022 are mainly centered on China, South Korea, Italy, and the Russian Federation, while Central and Eastern European countries like Poland and Turkey, Southeast Asian countries such as Singapore, Malaysia, and Indonesia, as well as African countries like South Africa also occupy prominent positions. In general, the trade network under the import perspective is relatively stable, and the total import trade is in a stage of steady growth (Fig 4). From 2000 to 2003, the lithium trade patterns that were initially formed in the countries along the path were formed, and there is not much of a trade volume difference between the countries. By 2008, the trade relationship among the countries along the route and the enhancement in trade intensity, the trade network of basic lithium products has basically been formed. Compared with 2008, the trade network of the countries across the route in 2013 was more intensive and complex, and the overall value of import trade had doubled, suggesting the gradual emergence of the "Belt and Road" cooperation initiative. According to the volume of import and export trade, no significant change in trade volume in 2019. While 2019 could have been impacted by COVID-19 outbreak, it was the end of the year and the impact on the economic and trade activities of the countries along the route was minimal. In 2022, despite the trade network appears to be sparse, lithium account for more than10 billion U.S. dollars of total imports. China imported lithium worth 6.7 billion U.S. dollars solely from Chile, constituting more than half of the whole trade volume of countries along the B&R with lithium resources. On the one hand, it demonstrates that the mutually beneficial and

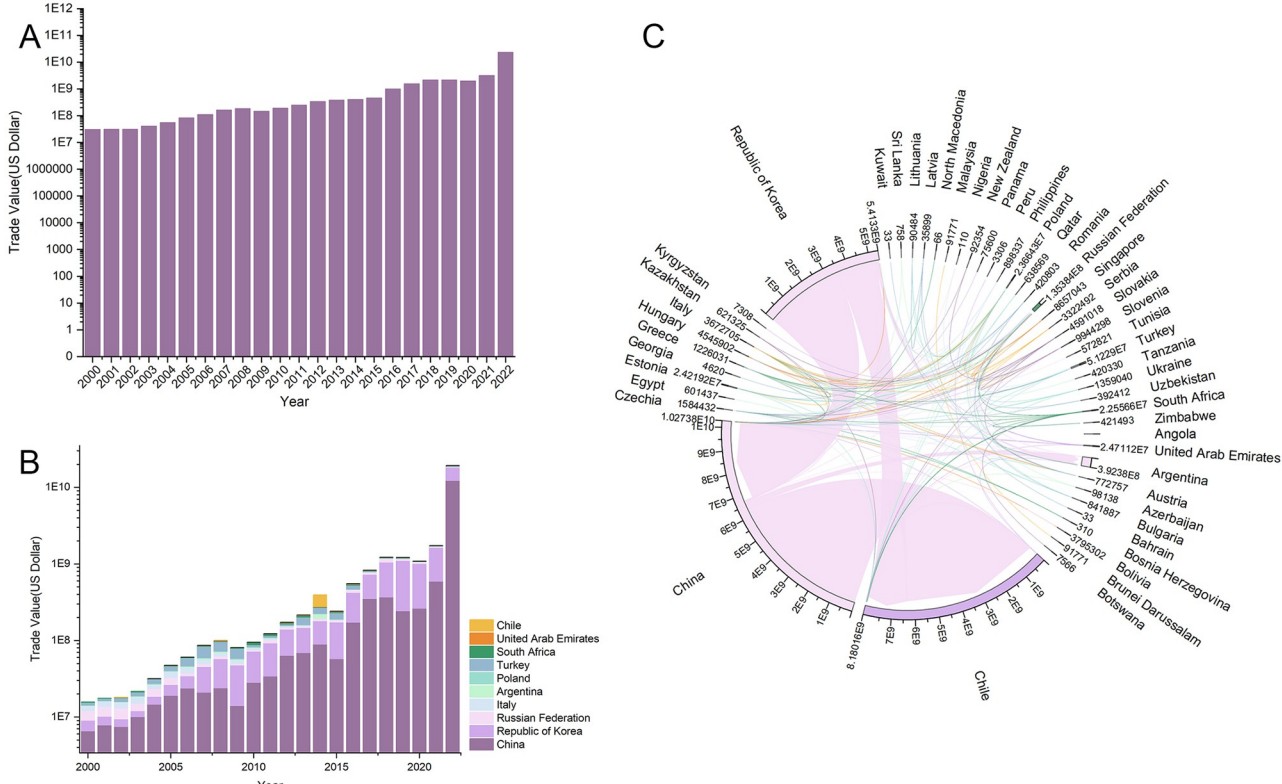

**Fig 4. Current status of the "Belt and Road" lithium trade, spanning the years 2000–2022.** (A) Total lithium trade of countries along the "Belt and Road", 2000–2022. (B) The principal "Belt and Road" lithium-importing nations from 2000 to 2022. (C) Lithium trade flows along the "Belt and Road" in 2022.

amiable collaboration between China and Chile within the framework of "One Belt, One Road" has achieved tangible achievements, and the extensive strategic partnership between the two countries has been in a state of constant expansion. On the other hand, with the booming development of China's new energy car manufacturing industry, lithium carbonate as the primary raw material for new energy automobile power battery, resulting in a sustained high demand for lithium resources domestically.

## 3.2 Analysis of whole export trade pattern

From an export perspective, lithium' export in countries within the B&R is relatively concentrated, and the exporting countries are primarily centralized in Chile, China, Italy and other places. As can be seen from Fig 3, the overall export pattern is stable, the export trade volume is also growing, but it is obvious that the number of countries trading with Russian Federation displays a fluctuating trend of decreasing, then increasing and then decreasing again. From 2000 to 2008, the export market of Russian Federation at this stage was first centered in Central and Eastern Europe and Central and East Asia, with export trade limited to only a few countries. However, the volume of exports significantly increased after 2013, and the expansion of export markets to Southeast Asia and Africa since 2019 highlights the growing international participation and influence of the BRI. In 2022, due to a lack of import and export data provided to the United Nations by Russian Federation as a reporting country, the data for that year was substituted with data from other countries along the same route. The total import and export trade for that year amounted to $240 million. Compared to the previous year, Russian Federation has experienced a notable decline in their total import and export trade of lithium in 2022, particularly with several Central and Eastern European countries. Meanwhile, Russian Federation's trade relations with China on basic lithium products have remained stable, which indicates that the scale and quality of China-Russia economic and trade cooperation is steadily improving, and that the drastic changes in the international landscape have prompted Russia to shift the center of gravity of its international commerce and economic ties with the Asia-Pacific area.

# 4 An analysis of the overall lithium trade network in the nations bordering the "Belt and Road"

## 4.1 Analysis of network density and reciprocity

Since 2000, the overall network density has gradually increased (Fig 5), but still at a low level. Due to the narrow range of commodities studied in this research and the large scope of the study area, the production and consumption markets of basic lithium products are largely concentrated in some countries with developed high-tech industries such as aerospace and new energy, and there are fewer trade links between marginal countries, so the density of the network has a low value. There are a lot of marginal countries in the network, the most of which are involved in unidirectional trade, as evidenced by the network's reciprocity, which is likewise erratic, growing, and at a low level. In conclusion, the network tends to develop closer and more stable, but the effects of the spiral expansion continue to negatively affect the network's stability because of the state of the global economy, pertinent institutional policies, and global public health problems.

## 4.2 Analysis of core periphery structure

This research computes the core and edge nations based on the Core Periphery in the software of UCINET and visualizes them using Gephi software (Fig 6) in an effort to more effectively

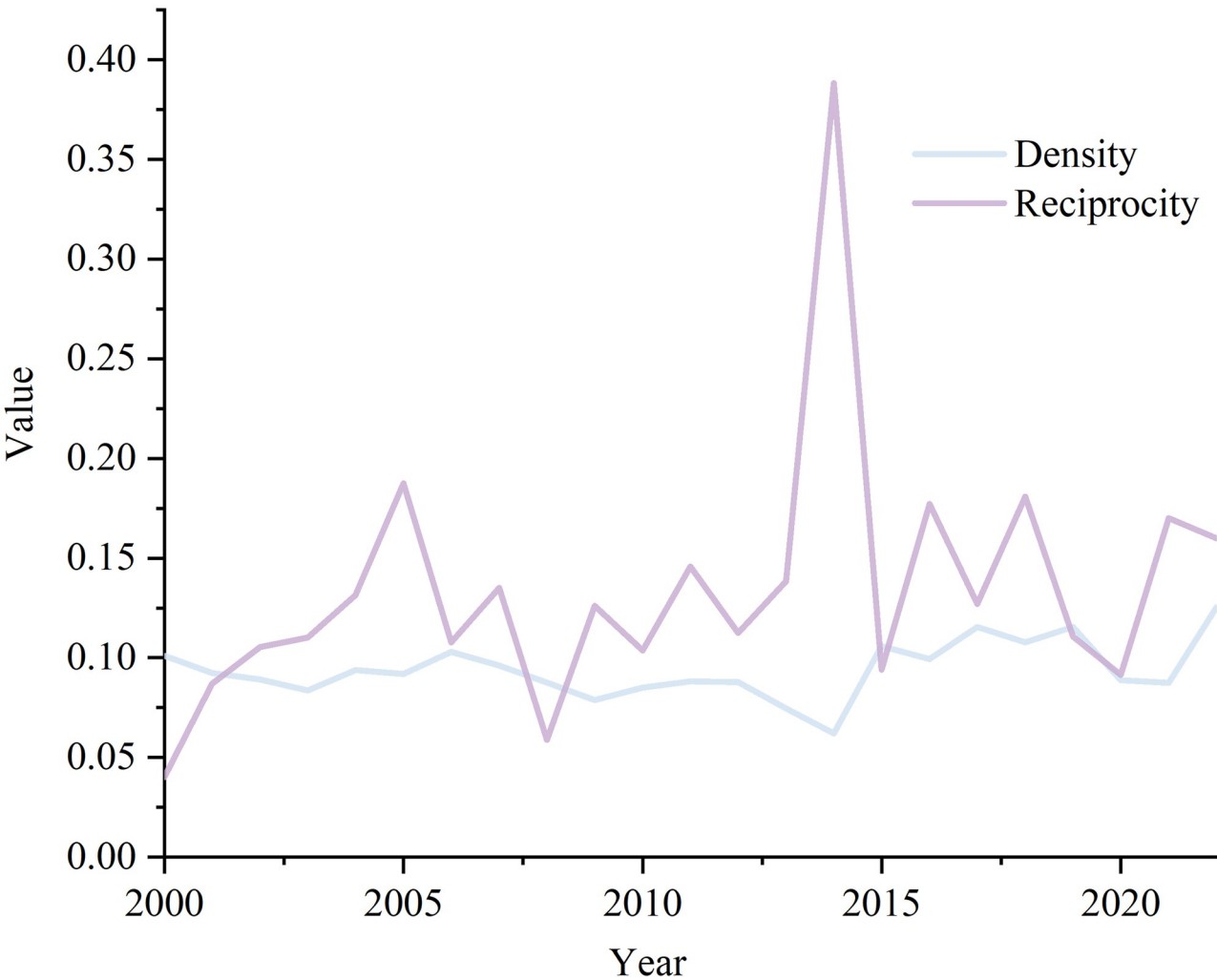

**Fig 5. "Belt and Road" trade network density value and reciprocity value.**

comprehend the hierarchical nature of the trade network of basic lithium goods of countries along the B&R. From 2000 to 2022, the number of core countries shows a decline trend, and the quantity of core countries decreases from 15 in 2000 to 10 in 2022. Argentina, Austria, Chile, China, the Czechia, Italy, South Korea, Poland, Russian Federation, and Slovenia have always remained in the core tier, and they are mainly parts of East Asia, South America, and Eastern Europe. However, in 2022, Southeast Asian countries have all withdrawn from the core layer, such as Indonesia, Malaysia, and Singapore, the COVID-19 pandemic has affected the tourism industries of Southeast Asian countries to varying degrees, which is their primary source of income. While this has seriously hampered these countries' ability to develop economically, there is still much space for improvement in terms of optimizing their industrial structures. The change in the number of core countries not only reveals the differences in resource endowment among countries along the route, but also the gradual increase in the gap between the level of economic development and the level of scientific and technological development is the principal reason for the change. By comparing the core-edge structure map from 2000 to 2022, the trade of basic lithium products before 2013 was mostly concentrated

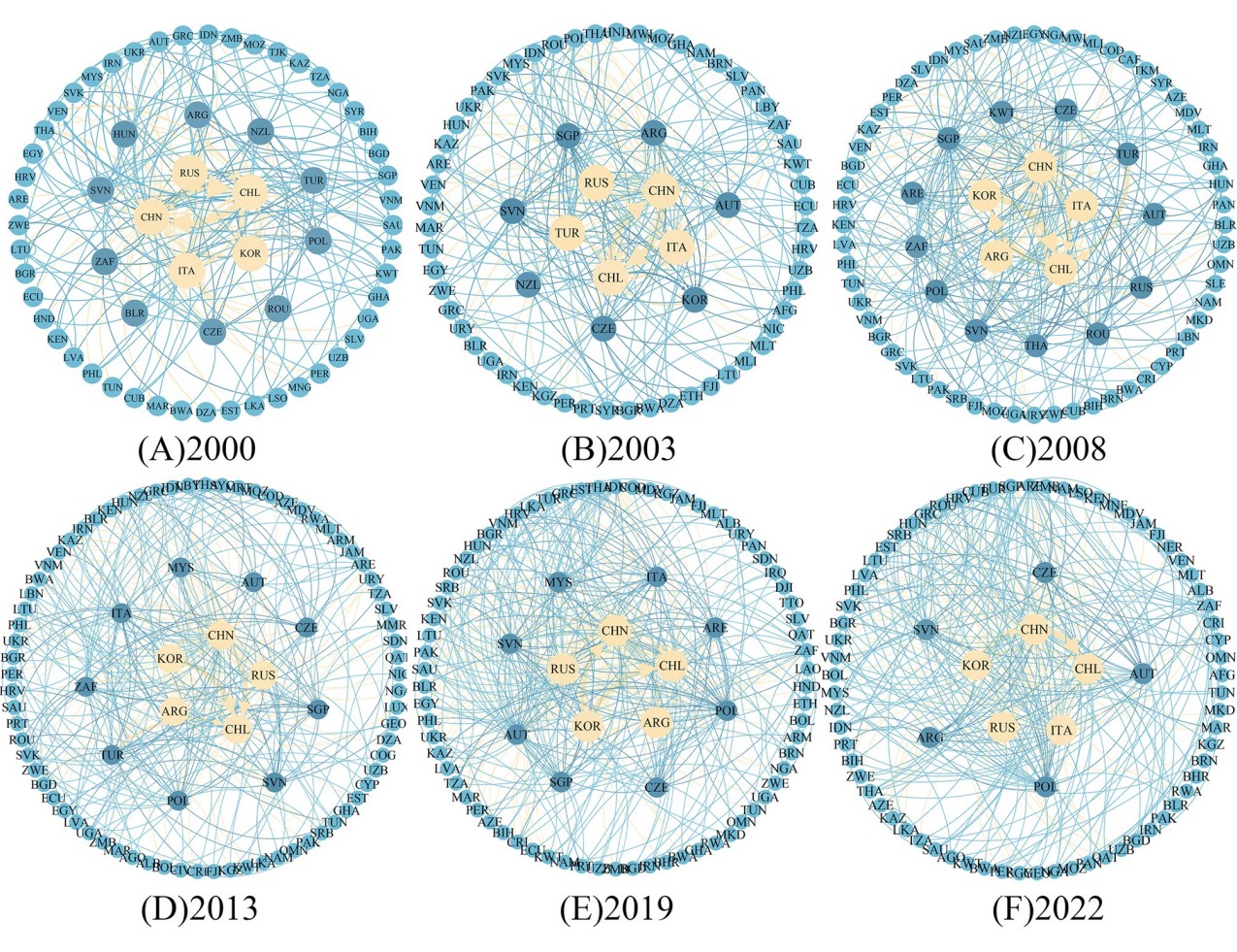

**Fig 6. Core periphery structure.**

between core countries, while the trade links between core nations, between core countries and edge countries, and between edge countries and edge countries in 2019 and 2022 have all increased in general. This indicates that the role and voice of the Belt and Road Cooperation Initiative on international platforms is prompting, which is more conducive to promoting trade liberalization and facilitation and has a positive impact on steadily advancing regional integration.

### 4.3 Block model analysis

Hierarchical clustering was performed through the CONCOR function in the UCINET software, selecting 2 and 0.2 as the maximum segmentation depth and convergence criteria, with a maximum of 25 iterations, and the network was divided into four blocks, as shown in Table 3.

In 2000, the first, second, third and fourth segments were the export trade sector, the two-way trade sector, the isolated marginal sector and the import trade sector, respectively. Among them, the first segment is mainly in Argentina, South Korea, Belarus and other South America, East Asia and Eastern Europe, and the proportion of the internal share of the whole is 42.05%, which is larger than the expected value of 4.26%, classifying it as the export trade segment. The

**Table 3. Characteristics of trade relations in each section of the trade network.**

| Year | Block | $I_k/I_{kt}$ | $I_{ke}/I_{ki}$ | $(n_k - 1)/(n - 1)$ | Classification |
|------|-------|--------------|-----------------|---------------------|----------------|
| 2000 | 1 | 42.05% | 15.00% | 4.26% | Export trade sector |
|      | 2 | 44.32% | 35.00% | 6.38% | Two-way trade sector |
|      | 3 | 7.95% | 3.33% | 0.00% | Isolated marginal sector |
|      | 4 | 5.68% | 46.67% | 8.51% | Import trade sector |
| 2003 | 1 | 15.38% | 68.57% | 3.77% | Export trade sector |
|      | 2 | 15.38% | 10.48% | 75.47% | Isolated marginal sector |
|      | 3 | 23.08% | 4.76% | 1.89% | Export trade sector |
|      | 4 | 46.15% | 16.19% | 56.60% | Isolated marginal sector |
| 2008 | 1 | 51.06% | 69.33% | 53.33% | Import trade sector |
|      | 2 | 19.15% | 2.67% | 46.67% | Isolated marginal sector |
|      | 3 | 27.66% | 18.67% | 25.00% | Export trade sector |
|      | 4 | 2.13% | 9.33% | 10.00% | Isolated marginal sector |
| 2013 | 1 | 65.06% | 60.81% | 37.14% | Two-way trade sector |
|      | 2 | 13.25% | 9.46% | 32.86% | Isolated marginal sector |
|      | 3 | 12.05% | 10.81% | 2.86% | Export trade sector |
|      | 4 | 9.64% | 18.92% | 24.29% | Isolated marginal sector |
| 2019 | 1 | 56.00% | 75.96% | 37.50% | Two-way trade sector |
|      | 2 | 24.00% | 19.23% | 39.06% | Isolated marginal sector |
|      | 3 | 20.00% | 4.81% | 14.06% | Export trade sector |
|      | 4 | 2.13% | 0.00% | 4.69% | Isolated marginal sector |
| 2022 | 1 | 51.49% | 33.66% | 34.62% | Two-way trade sector |
|      | 2 | 14.85% | 7.92% | 13.46% | Export trade sector |
|      | 3 | 30.69% | 29.70% | 26.92% | Two-way trade sector |
|      | 4 | 2.97% | 28.71% | 19.23% | Import trade sector |

segment is active in foreign export trade. The second segment includes 18 countries such as Russia, Italy, and Malaysia, mainly in Southeast Asia and Eastern Europe. The internal share of the overall is 44.32%, which is greater than the expected value of 6.38%, and the export trade is 35.00% of imports, which classifies it as a two-way trade segment. The third segment includes five countries, including Austria and Slovenia, and exports are 0.00% of imports, which is less than 1, so it is classified as an isolated marginal segment, with inactive trade to internal and external section. The fourth sector is 12 countries such as Chile, China, Kenya, etc., and the internal share of the overall share is 5.68%, which is smaller than the expected value of 8.51%, so it is classified as an import trade sector. The internal trade links of the sector are strong, and the external trade links are weak. From 2003 to 2022, after a long period of trade development, the division of internal plate varies greatly. This is due to the increasing number of countries participating in the B&R trade network, and the internal and external divisions' segments became gradual complicated. In 2019, the first sector is a bidirectional trade sector, which includes 25 countries such as Argentina, China, Chile, South Korea, Turkey, Singapore, etc., and the second, fourth sector is an isolated edge structure, totaling 40 countries. Two isolated edge sectors appear in both 2013 and 2019, mainly in Central Asia and Africa, such as Pakistan, Tanzania, and Zimbabwe. In 2022, two two-way trade sectors appear for the first time in that year, which have 20 and 15 countries respectively, mostly in the 2019 two-way trade section. Meanwhile, there is no isolated marginal section in that year, reflecting the gradual formation of the lithium trade pattern in the regions along the B&R.

**Table 4. Top 10 countries for basic lithium products centrality in 2022.**

| Degree Centrality | | Betweenness Centrality | | Closeness Centrality | |
|---|---|---|---|---|---|
| Nation | Index Value | Nation | Index Value | Nation | Index Value |
| China | 0.42 | China | 0.23 | Italy | 0.58 |
| Chile | 0.33 | Italy | 0.11 | China | 0.56 |
| Korea | 0.22 | South Africa | 0.07 | Poland | 0.52 |
| Argentina | 0.08 | Poland | 0.05 | Korea | 0.50 |
| Russia Federation | 0.08 | Chile | 0.05 | Slovenia | 0.50 |
| Turkey | 0.07 | Singapore | 0.04 | Czechia | 0.48 |
| United Arab Emirates | 0.07 | United Arab Emirates | 0.04 | Singapore | 0.48 |
| Estonia | 0.03 | Turkey | 0.04 | Austria | 0.45 |
| Poland | 0.03 | Korea | 0.03 | Greece | 0.45 |
| South Africa | 0.01 | Austria | 0.03 | United Arab Emirates | 0.44 |

## 4.4 Centrality analysis

Through the calculation of the three centrality indicators, the foremost ten nations that trade in terms of basic lithium product centrality in the B&R in 2022 are obtained, as shown in Table 4. As a whole, China, Italy, Poland, Austria, South Korea and the United Arab Emirates are listed in the foremost ten of the three centrality indexes in 2022, which fully proves that these six countries play a controlling role in the basic lithium products trade along the B&R, and are the controlling hubs in the basic lithium products trade network and have strong connectivity. They are the control hubs in the trade network of basic lithium products and have strong connectivity. Among them, China's degree centrality and betweenness centrality are ranked highest on the list with a significant advantage. At the same time, its closeness centrality is slightly weaker than that of Italy, which indicates that although China's foreign trade has weakened after the COVID-19 epidemic, In the nations along the line, it continues to be the hub of the commerce network, reflecting the role of the BRI in promoting the trade of lithium basic products. Chile and South Africa, both of which possess substantial lithium reserves, are located on the west coast of South America and in the south of Africa, respectively. Due to their considerable distance from other countries within the net, their connectivity and influence levels within the network are comparatively low. Slovenia, the Czechia and Greece in Central and Eastern Europe, Singapore in South Asian, and South Korea in East Asian all display notable centrality in the network. This reflects the trend of diverse countries participating in the "Belt and Road" trade, and the participation of countries with varying resource endowments and production advantages is also increasing. It is worth noting that the UN database lacks data on Russian Federation as a reporting country in 2022, their import and export trade volume has not declined much, and indicators such as centrality are difficult to measure, although this country has always ranked at the top of the list in the trade of basic lithium products in previous years. As we all know, in February 2022, the Russian-Ukrainian war broke out, and both Russia and Ukraine sanctioned and restricted trade with each other, and the war had a heavy impact on the economy and trade of the two countries.

## 4.5 PageRank analysis

For PageRank value analysis, the probability standard and acceptable error were determined to be 0.85 and 0.0001, respectively (Fig 7). The top two PR values are occupied by China and Chile, which is while indicating the importance of these two nations in the network, also

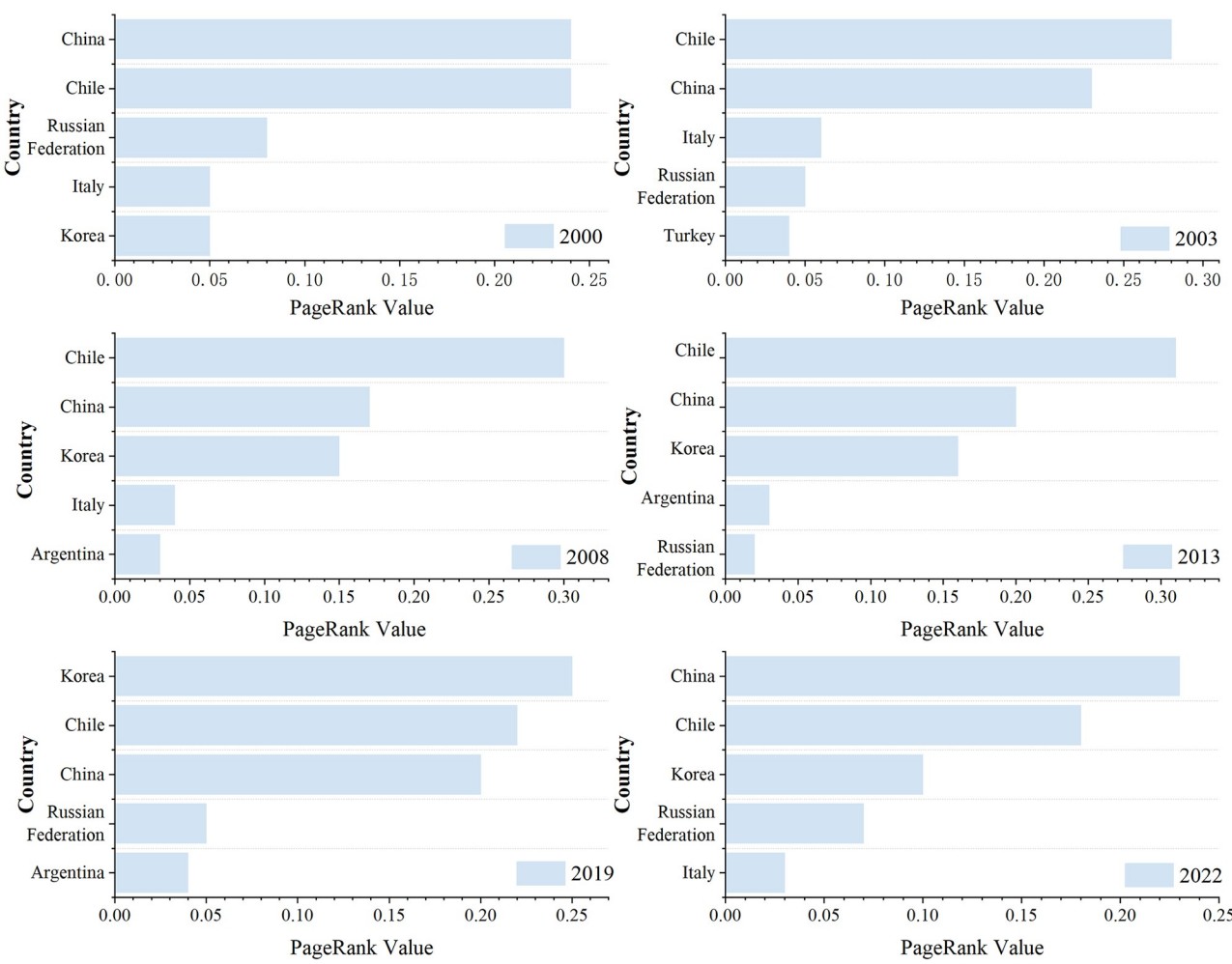

**Fig 7. Top five nations for lithium trade networks based on PageRank.**

reflects the fact that the PR value is dependent on the network's topology. Once the PR value is established, it shows a certain level of stability in the network connection relationship of basic lithium product trade over the 2000–2022 period. Crucial functions are also played by nodal nations including Argentina, Italy, South Korea, and the Russian Federation. The PR values for key nodes are computed differently from the centrality metrics, with Argentina's small centrality but important role as a resource-rich nation standing out in the network. Because of its distinct resource advantages, Chile's PR value ranked highest from 2000 to 2013 and consistently outperformed other nations along the "Belt and Road"; from 2013 to 2019, each country's overall PR value decreased slightly from 2013 to 2019, but it was distributed more evenly, possibly as a result of more nations engaging in the trade of lithium basic products. This could be because more nations are participating in international trade and economic activities along the route as a result of the proposal of cooperation initiatives and the increase in the number of countries trading basic lithium products. From 2019 to 2022, South Korea's PR value declined more than other countries due to its lack of mineral resources, strong reliance on the resources of other nations along the route, and reliance on the external market for the import of large amounts of energy and raw materials, which raises costs and risks for the country

while also placing it in a passive position and requiring energy security. This places Korea in a passive and reliant position with regard to energy security, in addition to raising economic costs and dangers. Because of this, resource-poor countries are significantly more vulnerable to the detrimental effects of "black swan" occurrences on their social, political, and economic stability than are resource-rich nations.

# 5 Analysis of factors influencing the trade network of basic lithium products in countries along the Belt and Road Initiative

## 5.1 Indicator selection and modeling

This study explores the factors affecting the lithium's trade pattern evolution in the B&R along five dimensions: level of economic development, institutional factors, level of science and technology, geography and cultural factors (Table 5).

(1) Level of economic development [38]. Given that the level of economic development will have a certain impact on the occurrence of international trade and the volume of trade, the matrix of the difference in the gross domestic product of each economy is selected as an unbiased variable. (2) Institutional factors [13, 38, 39]. National laws, institutions and trade relations have a direct impact on the development of trade between countries, and scholars are mainly based on the United Nations Global Governance Indicators to examine institutional differences. This paper selects three types of indicators to reflect its impact on trade, firstly, the average tariff rate of mineral products under the HS code, tariff rate is the most important expression of international trade policy, is the proportion of the tax amount calculated when taxing the object of taxation as stipulated in the Customs Tariff Rules, which can have a direct impact on the price of lithium imports and exports. Secondly, to examine the impact of political stability and regional trade agreements, political stability is the basis for trade, and regional trade agreements between the two countries to help strengthen the intensity of trade. Therefore, the matrix of regional trade agreements between economies, the absolute value matrix of the difference in political stability and the absolute value matrix of the difference in the average tax rate of mineral products are selected as unbiased variables. (3) Technology level [40]. Technological change in changing a country's energy structure at the same time will trigger the restructuring of the relationship between energy supply and demand, thus creating a new focus of energy geopolitical games. Therefore, this paper selects the patent difference matrix of each country as an unbiased variable. (4) Geographic factors [39]. Geographic distance and bordering is an important factor affecting the occurrence or not of international trade and trade intensity, the closer the geographic distance between two countries the lower the transportation cost will be, making the possibility of trade between the two countries greatly increased. (5) Cultural factors [35]. Cultural identity may lead to differences in consumer

**Table 5. Definition of indicators and data sources.**

| Factor | Norm | Definition | Source |
|---|---|---|---|
| Economics | GDP | Matrix of differences in levels of economic development among economies | World Bank |
| System | RTA | Matrix of regional trade agreements among economies | WTO |
| | AGI | Matrix of absolute values of political stability differentials between economies in the global governance index | WGI |
| | TAX | Matrix of absolute values of differences in average tax rates on mineral products between economies | WITS |
| Technology | PAT | Matrix of patent differentials between economies | World Bank |
| Geography | DIST | Matrix of spatial distances between the capitals of the economies | CEPII |
| | CONTIG | Matrix of bordering economies | CEPII |
| Culture | CL | Matrix of common languages among economies | CEPII |

brand choice and trade levels, and countries with similar cultural backgrounds are more likely to be on the same plate.

Based on the above five dimensions, a model of the influencing factors of B&R's lithium trade network is constructed as follows:

$$W(t)^{s(d)} = f(GDP, RTA, AGI, TAX, PAT, DIST, CONTIG, CL) \qquad (7)$$

Where: $W(t)^{s(d)}$ is the weighted (s) and unweighted (d) matrix of lithium in the nations along the B&R in year t. The matrix of whole import trade of lithium between the countries along the B&R is used as the biased variable in the weighted regression, which means that the weighted and undirected trade network, and the 0–1 matrix of the trade relations is obtained as the unweighted variable in the unweighted regression, that is, the unweighted and undirected trade network after processing; GDP denotes the matrix of differences in the level of economic development between economies; RTA signifies the matrix of regional trade agreements between economies; AGI represents the matrix of absolute values of differences in political stability between economies in the Global Governance Index; TAX means the matrix of absolute values of differences in the average tax rate on mineral products between economies; PAT denotes the matrix of differences in patents between economies; DIST means the matrix of the spatial distances between the capitals of the economies; CONTIG represents the matrix of whether or not an economy is bordered by a border; and CL represents the matrix of common languages between economies.

## 5.2 QAP Correlation analysis

Based on UCINET software, 5000 random permutations of the matrices were performed, and the correlation analysis of each unbiased variable matrix with weighted trade and unweighted trade network via weighted undirected and unweighted undirected trade network relationship matrices and each unbiased variable matrix QAP correlation analysis results were obtained, where the level of significance passes the test of significance (Tables 6 and 7). In the weighted and unweighted networks, there are positive correlations between trade network correlations and the matrix of GDP differences, the matrix of regional trade agreements, the matrix of

**Table 6. Weighted network QAP correlation analysis results.**

| Model | Indicator | 2000 | 2003 | 2008 | 2013 | 2019 | 2022 |
|---|---|---|---|---|---|---|---|
| Weighted network correlation analysis | GDP | 0.029 (0.192) | 0.068** (0.040) | 0.083** (0.012) | 0.085*** (0.003) | 0.038* (0.086) | 0.079** (0.029) |
| | RTA | 0.051* (0.065) | 0.123*** (0.000) | 0.192*** (0.000) | 0.042*** (0.000) | 0.169*** (0.000) | 0.180*** (0.000) |
| | AGI | -0.030 (0.173) | -0.044* (0.052) | -0.032 (0.141) | -0.043** (0.037) | -0.033 (0.285) | -0.048* (0.071) |
| | TAX | 0.069*** (0.008) | 0.002 (0.388) | 0.058** (0.025) | 0.041* (0.055) | 0.064* (0.100) | |
| | PAT | 0.005 (0.352) | 0.084** (0.024) | 0.051** (0.045) | 0.083*** (0.004) | 0.028 (0.106) | |
| | DIST | -0.102*** (0.000) | -0.096*** (0.000) | -0.128*** (0.000) | -0.115*** (0.004) | -0.133*** (0.000) | -0.118*** (0.000) |
| | CONTIG | 0.186*** (0.000) | 0.223*** (0.000) | 0.237*** (0.000) | 0.015 (0.206) | 0.036* (0.091) | 0.014 (0.255) |
| | CL | 0.064** (0.029) | 0.088*** (0.002) | 0.152*** (0.000) | 0.104*** (0.000) | 0.111*** (0.000) | 0.068** (0.025) |

Note: The coefficients denoted by ***, **, and * are significant at the 1%, 5%, and 10% significance levels, respectively.

**Table 7. Unweighted network QAP correlation analysis results.**

| Model | Indicator | 2000 | 2003 | 2008 | 2013 | 2019 | 2022 |
|---|---|---|---|---|---|---|---|
| Unweighted network correlation analysis | GDP | 0.233*** (0.001) | 0.274*** (0.000) | 0.247*** (0.000) | 0.299*** (0.000) | 0.300*** (0.000) | 0.266*** (0.000) |
| | RTA | 0.062** (0.034) | 0.087*** (0.005) | 0.147*** (0.000) | 0.097*** (0.000) | 0.138*** (0.000) | 0.143*** (0.000) |
| | AGI | -0.023 (0.141) | -0.041** (0.018) | 0.012*** (0.253) | -0.040** (0.065) | 0.052*** (0.038) | 0.077*** (0.009) |
| | TAX | 0.070*** (0.006) | 0.003 (0.419) | 0.068*** (0.000) | 0.095*** (0.000) | 0.087*** (0.001) | |
| | PAT | 0.187** (0.022) | 0.248*** (0.004) | 0.204*** (0.003) | 0.293*** (0.001) | 0.292*** (0.001) | |
| | DIST | 0.092*** (0.006) | -0.046 (0.110) | -0.052** (0.049) | -0.036 (0.239) | -0.021 (0.393) | -0.044 (0.200) |
| | CONTIG | 0.208*** (0.000) | 0.225*** (0.000) | 0.177*** (0.000) | 0.012 (0.282) | -0.017* (0.246) | -0.029* (0.074) |
| | CL | 0.046* (0.093) | 0.053* (0.056) | 0.050** (0.034) | 0.035* (0.079) | 0.001 (0.474) | 0.025 (0.203) |

absolute values of differences in the average tax rates of mineral products, and the matrix of patent differences, while there are negative correlations with the matrix of absolute values of differences in the political stability of economies and the matrix of differences in the geographic distances between capital cities. To address the issue of multicollinearity among the parameters mentioned above, QAP regression was employed to examine the combined influence of the aforementioned unbiased variables on trade networks. By analyzing Tables 4 and 5, it is possible to draw the conclusion that the correlation coefficient value and significance level of each unbiased variable in the unweighted undirected trade network is better than that in the weighted undirected network in general. The reason is that the formation of trade relations can only produce the difference in trade intensity, which means that the creation of trade relations is a prerequisite for the difference in trade intensity. The trade intensity of basic lithium commodities among the countries along the B&R has a large difference, so the weighted undirected network matrix is excluded from the regression analysis.

## 5.3 QAP regression analysis

The findings of the unweighted and undirected regression analysis of the trade of basic lithium goods in the nations along the B&R were obtained using the MRQAP regression term in UCINET with 5000 permutations (Table 8). The eight variables that were chosen for this study each have varying degrees of impact on the network relationships that underpin the trade in basic lithium products.

Among the economic factors, the regression coefficients of GDP are positive and steady, and all of them have passed the test of significance level of 1%. This suggests that the difference in the economic level among the nations is advantageous to the trade of the countries along the route, and it is easier to establish the trade relationship of basic lithium products among the countries with large differences in the size of their economies. Large economic scale and high level of industrial development, the greater the demand for basic lithium products, it is simpler to build trade relations with countries with small economic scale and low industrial level.

Among the institutional factors, the coefficients of RTA increased after 2008 and passed the 1% significance level test, which has a greater influence in the network and tends to be stable. This indicates that the signature of RTAs between the two nations has a favorable impact on

**Table 8. Unweighted network QAP regression analysis results.**

| Model | Indicator | 2000 | 2003 | 2008 | 2013 | 2019 | 2022 |
|---|---|---|---|---|---|---|---|
| Unweighted network correlation analysis | GDP | 0.233*** (0.001) | 0.274*** (0.000) | 0.252*** (0.001) | 0.298*** (0.000) | 0.300*** (0.000) | 0.264*** (0.000) |
| | RTA | 0.015 (0.296) | 0.045* (0.052) | 0.146*** (0.000) | 0.097*** (0.001) | 0.138*** (0.000) | 0.140*** (0.000) |
| | AGI | -0.002 (0.450) | -0.015 (0.200) | -0.018 (0.251) | -0.039* (0.061) | -0.051** (0.042) | -0.074*** (0.008) |
| | TAX | 0.029 (0.109) | 0.001 (0.482) | 0.114*** (0.000) | 0.095*** (0.000) | 0.087*** (0.000) | |
| | PAT | 0.189** (0.021) | 0.245*** (0.002) | 0.219*** (0.002) | 0.292*** (0.001) | 0.291*** (0.001) | |
| | DIST | -0.037 (0.150) | -0.003 (0.463) | -0.052* (0.056) | -0.023 (0.248) | -0.013 (0.400) | -0.042 (0.206) |
| | CONTIG | 0.191*** (0.000) | 0.207*** (0.000) | 0.177*** (0.000) | 0.012 (0.226) | -0.017 (0.221) | -0.029** (0.044) |
| | CL | 0.028 (0.158) | 0.043* (0.052) | 0.050** (0.023) | 0.035* (0.066) | 0.001 (0.428) | 0.023 (0.195) |

the evolution of the trade network pattern, and the signing of RTAs is conducive to the establishment of inter-country trade relations in basic lithium products. The coefficient of TAX shows a significant positive correlation, but decreases year by year, and the regression coefficient of 2003 is very small and insignificant because of incomplete tariff data of that year. Over time, the number of countries participating in the Belt and Road Cooperation Initiative and signing common regional trade agreements increases, tariff preferences between countries are greater, and trade costs between trading countries decrease. The higher the degree of trade reciprocity, the more conducive to the generation of basic lithium products trade relations and enhance their trade intensity. The coefficient of AGI is small, showing a significant negative correlation, indicating that the impact on the trade of the countries along the route is small, which indicates that the worse the political stability is, the more unfavorable it is to carry out inter-country trade activities, and that a good political environment is a prerequisite for economic development.

Among the scientific and technological factors, PAT has a significant positive impact on the basic lithium products trade with nations along the Belt and Road. Basic lithium products belong to the middle reaches of the lithium industry chain, the processing and refining level of the country has certain requirements, and the level of refining technology has a direct impact on the output of basic lithium products. Therefore, the smaller the gap between the scientific and technological level of the country is easier to establish trade relations.

Among the geographic factors, the regression coefficients for DIST are all negative, with only 2008 passing the 10% significance level test, suggesting that the trade relationship for basic lithium products is little affected by geographic distance. The regression coefficients for CONTIG are positive for the years 2000–2013 and negative for the years 2019–2022, suggesting that the presence or absence of a border between countries produces a different effect in different time periods. On the one hand, this is due to the decentralized distribution of lithium resources, with Chile and Argentina being the main suppliers of the products in the trade network, while these two countries are distant from the rest of the network. On the other hand, the development and innovation of today's means of transportation have greatly reduced the temporal and spatial distances and the impact of geographical distance on regional economic activities.

Among the cultural factors, the regression coefficients of CL are all positive but not significant, and the common official language as a whole is positively impacted by the generation of trade networks for basic lithium products. Language differences are part of the cost of foreign trade between countries, and the use of the common official language facilitates inter-country trade exchanges and contributes to the reduction of inter-country trade costs. At the same time, the historical and cultural foundations of these countries and the values of their inhabitants are also similar, which is a solid foundation for related trade between countries [39].

# 6 Conclusion, suggestion and prospect

## 6.1 Conclusion

This study constructs the import and export trade network of basic lithium products on the basis of trade data from 152 countries throughout the Belt and Road from 2000 to 2022, and applies the social network analysis technique to analyze the characteristics of the pattern evolution and the factors influencing it:

(1) The import and export trade network pattern of basic lithium commodities in the nations along the "Belt and Road" has a certain path dependence, and the importing countries are also exporting countries, with the import and export markets primarily located in East Asia, Central and Eastern Europe, South America and Southeast Asia. From the import pattern, the import pattern of basic lithium products in countries along the route is highly concentrated, with obvious path dependence, and the import markets are mainly China, South Korea, Russia, etc. From the export pattern, the export pattern of basic lithium products in countries along the Belt and Road is stable, and the export markets are mainly concentrated in Chile and China.

(2) Key node countries such as China, Chile, South Korea, Russia and Argentina are at the center of the trade network of basic lithium products of the countries along the Belt and Road, and at the same time, they are closely linked together. Italy, Poland, Singapore, South Africa and other countries are at important nodes with strong connectivity.

(3) From 2000 to 2022, the density of the network as a whole steadily increases, and the trade policy implemented by the Belt and Road Initiative is clearly effective. The reciprocity of this network is also fluctuating and increasing with a large amplitude, unstable and at a low level, affected by the international economic situation, relevant institutional policies and global public health emergencies and other factors, the stability of the network continues to suffer from the impact of the spiral growth.

(4) The core periphery structure of the commerce network of basic lithium products in the countries within the Belt and Road is relatively stable, with the quantity of core countries decreasing with each year, which indicates that an increasing number of core countries facilitate trade. Before 2013, the trade of basic lithium products is mostly concentrated between core countries, while in 2019 and 2022, the trade links between core countries, between core countries and edge countries, and between edge countries and edge countries generally increase. Trade linkages between core countries, between core and fringe countries, and between fringe and fringe countries increase overall. Trade links between countries along the route need to be strengthened, and China is already at the absolute center of the trade network along the Belt and Road.

(5) When it comes to determining factors, the extent of economic and technological development among the countries along the Belt and Road has a active effect on the establishment of trade relations for basic lithium products; whether or not regional trade agreements are signed and the difference in the average tax rate for mineral products have a substantial beneficial effect on the formation of inter-country trade relations as well as the intensity of trade; and

the political stability of the countries, whether or not they are bordering countries, and the similarity in language and culture are all conducive to the establishment of close trade relations among the countries.

## 6.2 Suggestions

**6.2.1 Suggestions for Belt and Road Initiative country (China).** (1) Grasp the development opportunities of the Belt and Road Initiative, China should actively discuss and promote the establishment of a strategic cooperation mechanism for lithium resources in countries along the Belt and Road Initiative. The lithium resources governance system is in the early stages of development, with no governance core and authority, and a lack of global coordination mechanisms. China should actively promote the coordination of interests between resource countries and consumer countries, establish lithium industry chain organizations, promote the formation of a global consensus, and put forward a global program for the sustainable supply of green lithium resources.

(2) China should strengthen the degree of domestic lithium resources exploration and development, strengthen the technological innovation of lithium extraction from salt lakes, and increase the investment in the search for lithium salt lakes, lithium pyroxene and lithium mica. Meanwhile, we should also actively build a safe, stable and diversified supply system of lithium resources along the Belt and Road. Strengthen diplomatic relations with the South American "lithium triangle", enhance the level of strategic cooperation, the use of high-level diplomatic means to create a favorable international investment environment, technology and industrial investment in exchange for local lithium resources. Enhance China's lithium resources in the domestic and international double cycle the safety and security capacity of China's lithium resources in the domestic and international double cycle.

(3) China should improve its foreign-related legal service system. Enterprises not only have to consider industrial, technological, financial and other aspects of risk factors, but also face the host country as well as other countries of the political and social risk of the test. In order to effectively protect the safety of private enterprises investment in overseas minerals, China needs to establish a sound international commercial tribunal and joint arbitration mechanism to provide a trustworthy and dependable dispute resolution platform for China's overseas investment entities. Meanwhile, it should help enterprises carry out compliance training for overseas investment and enhance the comprehensive competitiveness of Chinese enterprises.

**6.2.2 Suggestions for Belt and Road participating countries (other countries).** (1) Core countries such as Chile, Argentina and Russia should actively promote the extension of their export markets to Southeast Asia, South Asia, Eastern Europe and Africa, so as to promote the realization of mutual benefits and interconnection of the Belt and Road lithium trade network. Core countries and edge countries should maintain stable relations with the original trading countries on the basis of active development of trade with other countries, with more countries to negotiate free trade agreements, investment protection agreements, reduce trade costs and improve the stability of the network.

(2) Belt and Road countries need to work together with the initiative country (China) to study the creation of rules, enhance the Belt and Road Initiative's global issue-setting and rule-making capabilities, and promote the transformation of the Belt and Road Initiative from commodity-factor flow-type opening to rule system-type opening. The co-construction countries will strengthen the construction of multilateral cooperation platforms for lithium resources, and enhance inter-regional cooperation around the Middle East, Asia-Pacific, Russia-Central Asia, Latin America, Africa and other regions according to their resource endowments.

(3) Actively participate in the construction of "Six Corridors, Six Roads, Multiple Countries and Multiple Ports" to improve the logistics accessibility of the countries along the routes, and build a modern comprehensive transportation system. Poland and Hungary, as the important gateways for China-EU liner to enter the European Union, should grasp the role as important nodes in the transportation network of the Belt and Road, and deepen economic and trade cooperation with China and other Asian countries. As an important carrier of the "Belt and Road" connectivity, China-EU liner maintains the stability and smoothness of the international industrial chain and supply chain. And also forges a bridge and link for mutual benefit and win-win situation of the countries along the route. Therefore, countries along the route should work together to promote the construction of regional lithium trade corridors, and strive to realize the effective connection of railroad and highway logistics networks of the countries along the route, and positively build an all-round, multi-level and composite logistics network of the countries along the route.

### 6.3 Prospect

This study has certain limitations, including the subjective selection of the influencing factors and the absence of a thorough analysis of the stability and resilience of the lithium industry chain from a global perspective. As a result, the study is unable to fully illustrate the factors influencing the basic lithium product trade network. Going forward, it will be imperative to develop a more rational research technique and theoretical framework that integrates both quantitative and qualitative approach to investigate the resilience of the lithium industry chain in a more thorough and globalized manner. The global economic recovery is currently under strain, financial market risks are on the rise, geopolitical conflicts have not abated, and global trade growth is facing numerous downside risks. These factors have a significant impact on the development of global strategic emerging industries, and it is urgent to conduct pertinent research on other key mineral resources that are essential to the growth of these industries.

## Supporting information

**S1 File. Original data, indicator data and research results.**
(ZIP)

## Author Contributions

**Conceptualization:** Ziyun Ruan, Peng Du.

**Data curation:** Ziyun Ruan, Yanan Jiao.

**Formal analysis:** Peng Du.

**Funding acquisition:** Peng Du.

**Methodology:** Ziyun Ruan, Yanan Jiao.

**Software:** Ziyun Ruan, Yanan Jiao.

**Supervision:** Peng Du.

**Writing – original draft:** Ziyun Ruan.

**Writing – review & editing:** Peng Du, Yanan Jiao.

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
