## [Decision Letter · Decision Letter 0]

8 May 2024

PONE-D-24-10189Analysis of Lithium Trade Patterns and Influencing Factors in the Regions Along the "Belt and Road"PLOS ONE

Dear Dr. Du,

Thank you for submitting your manuscript to PLOS ONE. After careful consideration, we feel that it has merit but does not fully meet PLOS ONE’s publication criteria as it currently stands. Therefore, we invite you to submit a revised version of the manuscript that addresses the points raised during the review process.

In order to improve the current manuscript, it is crucial to thoughtfully review and respond to the feedback provided by all reviewers. By carefully analyzing their suggestions and making required changes, you can greatly enhance the quality and impact of your work. Each reviewer's comments provide valuable perspectives that, if integrated effectively, can enhance the depth and strength of your research findings. It is therefore vital to allocate adequate time and effort towards addressing these issues to refine the manuscript to its maximum capacity.

We look forward to receiving your revised manuscript.

Kind regards,

Suhairul Hashim, PhD

Academic Editor

PLOS ONE

Journal Requirements:

"The authors are very grateful for the help of Assistant Professor Peng Du and the support of the National Natural Science Foundation of China [41701123]."

4. Please ensure that you refer to Figure 1 in your text as, if accepted, production will need this reference to link the reader to the figure.

**Additional Editor Comments:**

It's crucial that the authors thoroughly address all concerns raised by the reviewers to ensure the integrity and quality of their work. Addressing these concerns not only strengthens the credibility of the research but also demonstrates a commitment to transparency and rigor in the scientific process. By carefully considering and responding to each reviewer's feedback, the authors can enhance the clarity, validity, and overall impact of their study.

Reviewers' comments:

Reviewer's Responses to Questions

**Comments to the Author**

1. Is the manuscript technically sound, and do the data support the conclusions?

Reviewer #1: No

Reviewer #2: Yes

Reviewer #3: Yes

2. Has the statistical analysis been performed appropriately and rigorously? 

Reviewer #1: No

Reviewer #2: Yes

Reviewer #3: Yes

3. Have the authors made all data underlying the findings in their manuscript fully available?

Reviewer #1: No

Reviewer #2: Yes

Reviewer #3: Yes

4. Is the manuscript presented in an intelligible fashion and written in standard English?

Reviewer #1: No

Reviewer #2: Yes

Reviewer #3: Yes

5. Review Comments to the Author

Reviewer #1: This study studies the trade patterns and influencing factors of lithium in the regions along the "Belt and Road". The structure of the article is clear and logical. However, here are some comments for the author's reference:

1. Some sentences in the abstract are lengthy, which can be further simplified to improve readability.

2. The conclusion section of the abstract should more clearly summarize the main findings and implications of the study. While the current abstract lists the findings, it does not generalize about the implications or implications of the study in a whole.

3. Authors have included literature review in ‘Introduction’ section related to certain aspects of topic, but it does not seem to have a logical overall flow. Relevant literature review should be done in the line with the following question: What is the limitation about previous studies, and how this paper fills the gap? Your literature review should lead to the specific hypothesis for what you expect to find in targeted area.

4. Section data is used in part 2.2, which lacks time series information, so it cannot reveal the long-term trend and dynamic changes of lithium trade.

5. The empirical part can enrich some methods, such as principal component analysis and factor analysis. Further analysis of the basic lithium product trade network pattern of countries along the route.

6. Some specific measures can be proposed in the conclusion part.

7. Finally, the paper is poorly written and this current form requires serious improvement in all sections.

Reviewer #2: 1. Analysis of the evolution of trade import and export patterns should prioritize the overall spatial characteristics rather than merely discussing the network status of nodes and the strength of their connections. Additionally, assertions regarding changes in these patterns should be supported with literature or empirical evidence rather than being based on subjective speculation.

2. Analysis of structural features should focus on elucidating the relationships among different methodologies and the scientific questions they address, rather than merely supplementing the workload with classical social network analysis techniques.

3. Although numerous methods are employed throughout this paper to meet varied research objectives, it is advisable to consolidate these methods and present them in Chapter Two. This arrangement will clarify the interrelationships among the methods and their roles within the overarching research aims.

4.The selection of factors influencing the evolution of trade patterns in foundational lithium products under the "Belt and Road" initiative needs to be underpinned by additional literature to demonstrate their validity and scientific soundness.

5. The policy recommendations provided in the document should align more closely with the analytical results. The current analysis is overly generalized and lacks a strong connection to the substantive content of the research.

6. The introduction of the paper should be significantly enhanced with additional literature, addressing topics including but not limited to the global trade of lithium products and the application of social network analysis methodologies.

Reviewer #3: 1.What are the main indicators of social network analysis that need to be introduced in the Research Methodology section? What is the reason why the paper only selected some indicators? What is the reason why the author did not perform block modeling and core edge analysis?

2.The Research methodology section also needs to introduce the principles and advantages of QAP regression. Suggestions 5.1 and 5.2 are placed in the research methods section.

3. Section 3-4 needs to focus on the role of China and the "the Belt and Road" policy in the lithium trade network.

4. It is necessary to supplement the theoretical basis for the indicators selected in section 5.2.

5. Policy recommendations should be separately listed as a section of content. In addition, countermeasures and suggestions should be given to the Belt and Road Initiative (China) and participating countries (other countries) respectively.

6. PLOS authors have the option to publish the peer review history of their article (what does this mean?). If published, this will include your full peer review and any attached files.

Reviewer #1: No

Reviewer #2: No

Reviewer #3: No

---

## [Author Response · Author response to Decision Letter 0]

5 Jun 2024

Dear academic editor and reviewers:

We appreciate your comments and suggestions on our manuscript. These comments are valuable and helpful to improve our manuscript. According to the comments, we have considered your comments and made detailed revisions and improvements to this article. Due to the large number of changes, so the specific responses we put in the word documents.

Thank you and best regards.

---

## [Decision Letter · Decision Letter 1]

4 Jul 2024

Analysis of Lithium Trade Patterns and Influencing Factors in the Regions Along the "Belt and Road"

PONE-D-24-10189R1

Dear Dr. Du,

We’re pleased to inform you that your manuscript has been judged scientifically suitable for publication and will be formally accepted for publication once it meets all outstanding technical requirements.

Kind regards,

Suhairul Hashim, PhD

Academic Editor

PLOS ONE

Additional Editor Comments (optional):

Reviewers' comments:

Reviewer's Responses to Questions

**Comments to the Author**

1. If the authors have adequately addressed your comments raised in a previous round of review and you feel that this manuscript is now acceptable for publication, you may indicate that here to bypass the “Comments to the Author” section, enter your conflict of interest statement in the “Confidential to Editor” section, and submit your "Accept" recommendation.

Reviewer #1: All comments have been addressed

Reviewer #3: All comments have been addressed

2. Is the manuscript technically sound, and do the data support the conclusions?

Reviewer #1: Yes

Reviewer #3: Yes

3. Has the statistical analysis been performed appropriately and rigorously? 

Reviewer #1: Yes

Reviewer #3: Yes

4. Have the authors made all data underlying the findings in their manuscript fully available?

Reviewer #1: Yes

Reviewer #3: Yes

5. Is the manuscript presented in an intelligible fashion and written in standard English?

Reviewer #1: Yes

Reviewer #3: Yes

6. Review Comments to the Author

Reviewer #1: The detailed response to my last report is satisfactory, I thus recommend acceptance of the revised manuscript.

Reviewer #3: The authors revised this paper according to the comments of the reviewers, so it is recommended to accept it.

7. PLOS authors have the option to publish the peer review history of their article (what does this mean?). If published, this will include your full peer review and any attached files.

Reviewer #1: No

Reviewer #3: No

---

## [Editor Report · Acceptance letter]

10 Jul 2024

PONE-D-24-10189R1 

PLOS ONE

Dear Dr. Du, 

I'm pleased to inform you that your manuscript has been deemed suitable for publication in PLOS ONE. Congratulations! Your manuscript is now being handed over to our production team.

Kind regards, 

on behalf of

Dr. Suhairul Hashim 

Academic Editor

PLOS ONE